# Anatomical and Micro-CT measurement analysis of ocular volume and intraocular volume in adult Bama Miniature pigs, New Zealand rabbits, and Sprague-Dawley rats

Yajun Wu[1,2,3☉], Yuliang Feng[1,2,3☉], Jiasong Yang[1,2,3], Yuwen Ran[4], Zongtao Shu[5], Xiaobo Cen[5]*, Wensheng Li[1,2,3]*

1 Aier Academy of Ophthalmology, Central South University, Changsha, Hunan, China, 2 Department of Ophthalmology, Shanghai Aier eye hospital, Shanghai, China, 3 Shanghai Aier eye institute, Shanghai, China, 4 Changsha Aier eye hospital, Changsha, Hunan, China, 5 WestChina-Frontier PharmaTech Co., Ltd., Chengdu, Sichuan, China

☉ These authors contributed equally to this work.
* drlws@qq.com (WL); xbcen@scu.edu.cn (XC)

## Abstract

### Aim

Utilizing a combination of micro-computed tomography (micro-CT) and anatomical techniques for the volumetric assessment of the eyeball and its constituents in Bama Miniature Pigs, New Zealand rabbits, and Sprague-Dawley(SD) rats.

### Method

Six Bama Miniature pigs, New Zealand rabbits, and SD rats were enrolled in the study. Micro-CT and gross volumetric estimation of ocular volume were employed to acquire data on ocular volume, anterior chamber volume, lens volume, and vitreous cavity volume for each eye.

### Results

The eyeball volume of pigs ranges from approximately $5.36 \pm 0.27$ to $5.55 \pm 0.28$ ml, the lens volume from approximately $0.33 \pm 0.02$ to $0.37 \pm 0.06$ ml, the anterior chamber volume from approximately $0.19 \pm 0.05$ to $0.28 \pm 0.04$ ml, and the vitreous volume is approximately $3.20 \pm 0.18$ ml. For rabbits, the eye volume, lens volume, anterior chamber volume, and vitreous volume range from approximately $3.02 \pm 0.24$ to $3.04 \pm 0.24$ ml, $0.41 \pm 0.02$ to $0.44 \pm 0.02$ ml, $0.23 \pm 0.04$ to $0.26 \pm 0.05$ ml, and $1.54 \pm 0.14$ ml, respectively. In SD rats, the volumes are $0.14 \pm 0.02$ to $0.15 \pm 0.01$ ml for the eyeball, $0.03 \pm 0.00$ to $0.03 \pm 0.00$ ml for the lens, $0.01 \pm 0.00$ to $0.01 \pm 0.01$ ml for the anterior chamber, and $0.04 \pm 0.01$ ml for the vitreous volume.

**Data Availability Statement:** All relevant data are within the manuscript and its Supporting information files.

**Funding:** 2023 Research Fund of Aier Eye Research Institute(No.AEI202310LC01); Science Research Foundation of Aier Eye Hospital Group (No.AR2201D3). 1.2023 Research Fund of Aier Eye Research Institute (No.AEI202310LC01), Funder: Wensheng Li:Grant project funding, guide research design, and revise manuscripts. 2.Science Research Foundation of Aier Eye Hospital Group (No.AR2201D3), Funder: Jiasong Yang: Grant project funding, responsible for data checking and statistics.

**Competing interests:** NO authors have competing interests.

## Conclusion

The integration of micro-CT and gross volumetric estimation of ocular volume proves effective in determining the eyeball volume in Bama Miniature Pigs, New Zealand rabbits, and SD rats. Understanding the volume distinctions within the eyeballs and their components among these experimental animals can lay the groundwork for ophthalmology-related drug research.

## Introduction

The eyeball, a pivotal sensory organ with a precise refractive system, plays a crucial role in the human body [1]. Biomechanical processes involving fluid flow among different ocular structures are vital for delivering ophthalmic drugs [2]. Thus, elucidating the eyeball and its contents' volumes holds fundamental scientific value. This encompasses research on emmetropia and refractive errors mechanisms, establishing eye biological models, and contributes significantly to the diagnosis and treatment of various eye diseases.

Pigs, with eyes structurally and sizably similar to humans, are frequently employed as animal models for eye disease research [3–5]. Notably, studies using pigs, such as Chan et al. [6] have used pig eyes to validate the effectiveness of gene therapy. In addition, New Zealand rabbits, known for their docile nature and large eyes, are also prevalent in ophthalmic research [7, 8]. Sun et al. [9] utilized them for microwave thermokeratoplasty experiments. Besides, Rabbits, characterized by larger eyeballs and lacrimal glands akin to humans, are widely utilized in dry eye studies due to their stability [10]. Moreover, rabbits have large eyes that are conducive to the biochemical characteristics of drug research, and their usage cost is relatively low, though non-human primates are more similar to humans, their usage cost is very high, making rabbits a more commonly used choice for dry eye experiments [11]. Additionally, Sprague-Dawley(SD) rats, cost-effective with rapid reproduction, share a comparable eyeball structure with humans. Guo, et al., etc [12]. successfully established an animal model of diabetes by intraperitoneal injection of streptozotocin (STZ) into SD rats, and observed abnormal meibomian gland function in model rats, revealing that diabetes can cause dysfunction of the meibomian gland [12]. In conclusion, experimental animals, including Bama miniature pigs, New Zealand rabbits, and SD rats, are pivotal in advancing our understanding of human eye diseases in ophthalmology research.

Computed tomography (CT) imaging provides detailed visualization of the eyeball, offering high resolution and measurement accuracy. Qualitative and quantitative assessments of specific data can be conducted through three-dimensional reconstruction [13, 14]. Micro-CT, a miniature CT with micrometer-level resolution, enhances spatial resolution for imaging small animal disease models [15]. While Magnetic Resonance Imaging (MRI) excels in finer soft tissue imaging, its higher cost and longer scanning time, coupled with limitations in scanning metallic or paramagnetic tissues [16], especially in animal experiments, few laboratories have the conditions to equip Micro-MRI systems. Moreover, research has consistently demonstrated CT's efficacy in measuring ocular volume parameters. In fact, as early as 1984, one study used CT to measure the ocular volume of the human body to study its correlation with age and gender [17]. Besides, Ozer CM, et al. [18] also clearly indicated the important value of CT in measuring eyeball volume parameters. Also, CT is used in animal experiments, Leszczyński B, et al. [14] performed CT scans on the eyeballs of domestic pigs and combined staining methods to accurately obtain structurally clear pig eye tissue. Similarly, Salguero et al. [19]

measured the volume, density, and the length of the normal dog eye structures (Including the average axial length of the sphere, the average anterior posterior distance of the anterior chamber, the average anterior posterior distance of the vitreous chamber, etc) with CT.

Despite these advancements, uncertainties persist regarding specific parameters of animal eyeballs and whether imaging data align with actual values. In this study, we utilized micro-CT combined with anatomical methods to simultaneously measure ocular volume and intraocular volume in three common ophthalmic experimental species-Bama miniature pigs, New Zealand rabbits, and SD rats. This approach aims to obtain more accurate parameters for the eyeball and intraocular volume in these species, providing crucial reference values for related ophthalmic experiments.

## Method

### Ethics

All animals in this study were approved by the Institutional Animal Care and Use Committee (IACUC) of WestChina-Frontier Pharma Tech Co., Ltd. (Approval Number: IACUC-SW-S2023031-P001-01). Our experiment comply with the following animal research guidelines: Animal Research: Reporting in Vivo Experiments (ARRIVE) guidelinesthe, Association for Research in Vision and Ophthalmology (ARVO), and the American Veterinary Medical Association (AVMA) Guidelines for the Europe of Animals (2020).

### Experimental animals

We selected 6 healthy Bama miniature pigs (ordinary grade), aged 10–12 months, weighing 28–33 kg, with an equal gender distribution. The pigs were obtained from Chengdu Dashuo Experimental Animal Co., Ltd. (License: SCXK [Chuan] 2019–031). Additionally, 6 SPF grade SD rats (half male, half female), aged 2–3 months, weighing 330-420g, were purchased from Zhejiang Weitong Lihua Experimental Animal Technology Co., Ltd. (License: SCXK [Zhe] 2019–0001). The cohort also included 6 SPF grade New Zealand rabbits (half male, half female), 7 months old, weighing 3.0–3.5kg, sourced from Shandong Benming Biotechnology Co., Ltd. (License: SCXK [Lu] 2017–003). Each species of animal includes a total of 12 eyeballs. All animals were housed at WestChina-Frontier Pharma Tech Co., Ltd. (License for experimental animal use: SYXK [Chuan] 2021–238). The animals were maintained in a controlled environment with a temperature range of 16–26˚C for rabbits and pigs and 20–26˚C for rats (daily temperature difference $\leq$ 4˚C), relative humidity of 40–70%, and a 12/12-hour light/dark cycle.

### Micro-CT scanning and data analysis

We utilized the NEMO$^{®}$ Micro-CT (model NMC-100 by Life Medical Technology, Kunshan, China) for scanning. Scanning parameters were set as follows: large detached cabin for scanning; current of 0.2mA; voltage of 70kV; 40 frames per second; 360 frames per turn; and a scanned pixel size of 0.03*0.03*0.03mm. Reconstruction parameters were configured with an iterative algorithm, a reconstruction separation rate of 1k*1k, and a reconstruction pixel size of 0.015*0.015*0.015mm.

Volumetric measurements and image analysis:We employed Avatar software (version 1.6.6.7, provided by Pingsheng Medical Technology, Kunshan, China) for image analysis. The CT reconstructed image was imported into Avatar. Using the [Hand-drawn Mask] tool, the specific way we locate the eyeball for scanning is to ensure that the entire eyeball is located within the scanning interval, manually select the lens to start drawing from the tip of the ciliary

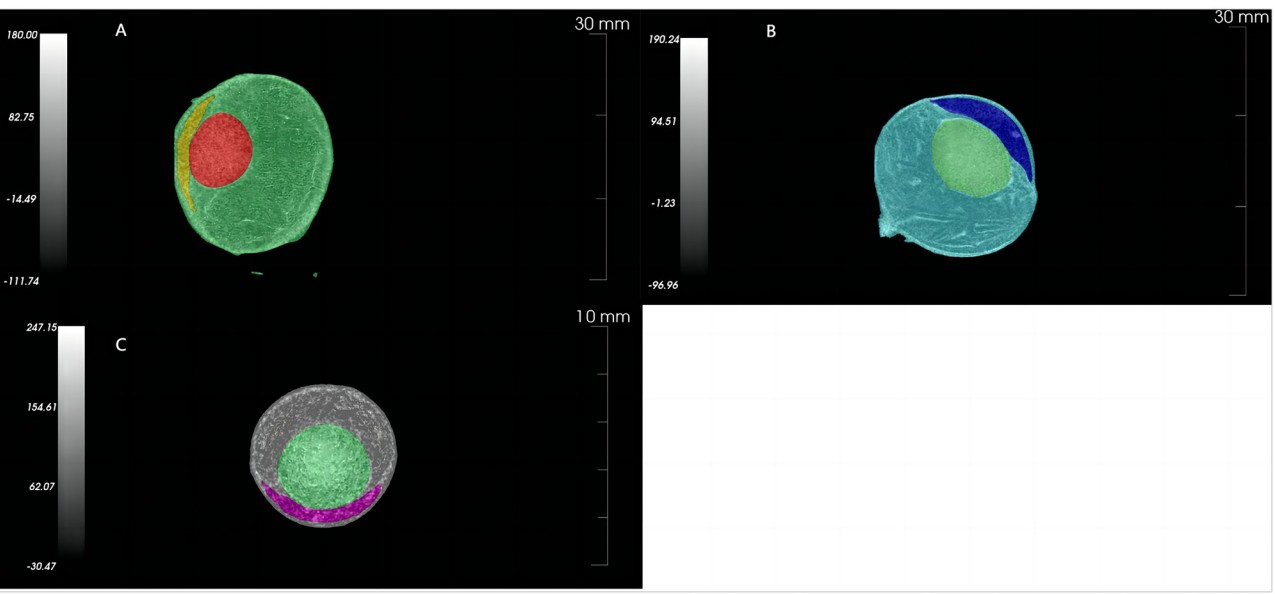

**Fig 1. CT images of the eyeballs and contents of different species of animals after manual delineation.** Note: A, CT images of the eyeballs and contents of adult Bama miniature pigs; B, CT images of the eyeballs and contents of adult New Zealand rabbits; C, CT images of the eyeballs and contents of adult SD rats.

body, move towards the other end until the lens structure disappears, and then use this tool to draw the region of interest (ROI) of the lens until the anatomical structure disappears; Using the same method, select any starting surface of the eyeball or anterior chamber structure and draw its ROI until the anatomical structure disappears, the vitreous body and posterior chamber cannot be located due to the lack of a clear boundary line. Each ROI can be divided into multiple sections when drawing, and the software automatically calculates the ROI volume of the middle section. After the ROI of each eye structure such as anterior chamber and lens was drawn, Avatar software provides mask of the lens and anterior chamber, and automatically calculates its volume as the physiological volume of the corresponding structure (see Fig 1), without secondary or manual calculation.

The total eyeball volume was determined using the threshold selection method. The vitreous body volume was calculated using Boolean operations based on the difference between the total eyeball volume and the lens and anterior chamber volume. It is important to note that the volume of the vitreous cavity obtained here represents the sum of the vitreous cavity and the remaining eye tissue, as the CT images lack clarity in delineating the vitreous cavity and posterior chamber. The actual volume of the vitreous cavity was measured using anatomical methods.

All operations were performed by a single experimenter (ZTS).

## Gross volumetric estimation of ocular volume methods

Pigs and rabbits: After intramuscular injection of 2mg/kg of diazepam (Sigma-Aldrich, USA), intravenous injection (via ear vein after skin preparation) of pentobarbital sodium (30mg/kg, Sigma-Aldrich, USA) was administered, and ultimately all animals were euthanized through abdominal aortic bleeding. Only when the animal's breathing, heartbeat, and toe reflexes completely disappear can it be confirmed that the animal has died, and rapid enucleation of the eyeball can be performed. Both eyeballs were removed and placed in a 10ml measuring

cup. Subsequently, 3ml of distilled water was added incrementally using 0.5ml, 0.3ml, and 0.1ml pipettes until reaching the 10ml mark. The total transferred water amount was recorded to determine the total ocular volume. A 1ml syringe was used to extract aqueous humor until the anterior chamber was dry, and the volume of the aqueous humor was recorded. After aqueous humor removal, the eyeball underwent rapid freezing in liquid nitrogen, and the complete lens and vitreous body were dissected. These tissues were then cut into pieces and placed into a 5ml measuring cylinder. Following tissue melting, 0.5ml, 0.3ml, and 0.1ml pipettes were used to inject distilled water into the measuring cup until reaching the 5ml mark. The total transferred water amount was recorded to obtain the volume of the vitreous body and lens.

Rats: The measurement method for the eyeball volume of SD rats is the same as that of pigs and rabbits. Unlike euthanasia and dissection of pigs and rabbits using intravenous anesthesia, SD rats were anesthetized with 30mg/kg pentobarbital sodium (Sigma-Aldrich, USA) via intraperitoneal injection, followed by bloodletting of the abdominal aorta and enucleation of the eyeball. Besides, unlike the 10ml measuring cup used for the other two types of eyeballs, the eyeballs of rats were placed in a 5ml measuring cup to determine ocular volume. A 0.3ml insulin syringe was used for extracting aqueous humor. The lens and vitreous body removal method remained the same as in rabbits and pigs. Extracted crystals and vitreous bodies were placed into the 0.1ml scale of the syringe for insulin, and the volume was read after tissue melting.

All the aforementioned procedures were conducted by a single experimenter.

## Statistical methods

Statistical analysis of the data was performed using SPSS 25.0 (IBM Corp., Armonk, NY, USA). All results are presented as mean±standard deviation (SD). A comparison of volumes measured by the two methods was carried out using unpaired t-tests, Pearson correlation analysis was performed using the software Origin (version 2022, Originlab, USA), and $p \leq 0.05$ indicated a statistically significant difference.

## Result

### Results of ocular volume and intraocular contents in Bama Miniature Pigs

No significant difference was observed between the anatomical measurement data and CT data of ocular volume and vitreous cavity volume in Bama miniature pigs ($p > 0.05$), as illustrated in Fig 2A and 2D. However, the gross volumetric estimation of ocular volume indicated greater lens volume and anterior chamber volume compared to CT results, as depicted in Fig 2B and 2C, respectively. In addition, the eyeball volume of pigs was about 5.36±0.27 to 5.55 ±0.28ml, the lens volume was about 0.33±0.02 to 0.37±0.06ml, and the anterior chamber volume was about 0.19±0.05 to 0.28±0.04ml. The vitreous volume was about 3.20±0.18ml. Refer to Table 1 for detailed data.

### Results of ocular volume and intraocular contents in New Zealand rabbits

Similar to the findings in Bama miniature pigs, there was no significant difference between the gross volumetric estimation of ocular volume and CT results for ocular volume and vitreous cavity volume in New Zealand rabbits ($p > 0.05$), as depicted in Fig 3A and 3D, respectively. However, the gross volumetric estimation of ocular volume of lens volume and anterior chamber volume were notably higher than the CT results, as illustrated in Fig 3B and 3C, respectively. The eyeball volume of rabbits was about 3.02±0.24 to 3.04±0.24ml, the lens volume was about 0.41±0.02 to 0.44±0.02ml, the anterior chamber volume was about 0.23±0.04 to 0.26

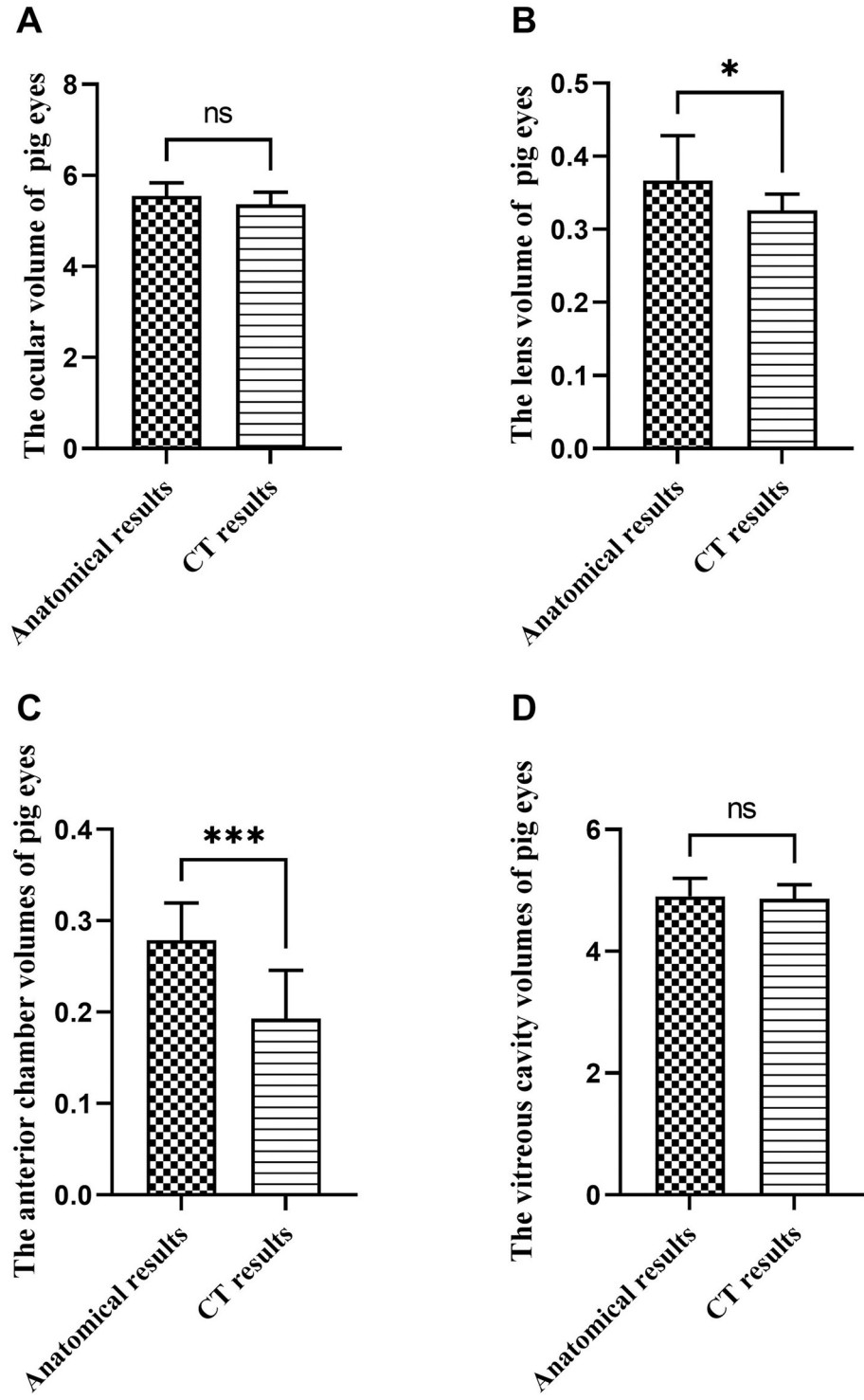

**Fig 2. Comparison of anatomical and CT methods for measuring the eyeball and intraocular volume in Bama miniature pigs (ml).** Note: A, Comparison of two measurement methods in pig ocular volume; B, Comparison of two measurement methods in pig lens volume; C, Comparison of two measurement methods on the volume of pig anterior chamber; D, Comparison of two measurement methods on the volume of pig vitreous cavity (The volume of the vitreous cavity here represents the total volume of the vitreous cavity and the remaining eye tissue). ns means that the difference is not statistically significant; *Indicates P ≤ 0.05; *** Indicates P ≤ 0.001.

**Table 1. Comparison of anatomical and CT measurements of ocular volumes and intraocular contents volumes in pigs (ml).**

| Group (n = 12) | Ocular volume | Lens volume | Anterior chamber volume | Vitreous cavity volume (volume of vitreous cavity and remaining eye tissue) | Real vitreous cavity volume |
|---|---|---|---|---|---|
| Gross volumetric estimation of ocular volume | 5.55±0.28 | 0.37±0.06 | 0.28±0.04 | 4.91±0.29 | 3.20±0.18 |
| CT result | 5.36±0.27 | 0.33±0.02 | 0.19±0.05 | 4.86±0.23 | / |
| *t* value | 1.66 | 2.15 | 4.45 | 0.40 | / |
| *P* value | 0.11 | 0.04* | ≤ 0.00*** | 0.69 | / |

Note:

*Indicates P ≤ 0.05;

*** Indicates P ≤ 0.001.

±0.05ml, and the vitreous volume was about 1.54±0.14ml. Detailed data can be found in Table 2.

### Results of ocular volume and intraocular contents in SD rats

There was no statistically significant difference (*p*>0.05) between the anatomical and CT measurements of eyeball volume and anterior chamber volume in SD rats, as demonstrated in Fig 4A and 4C. However, the CT measurements of lens volume were significantly larger than the gross volumetric estimation of ocular volume, as depicted in Fig 4B. Conversely, the gross volumetric estimation of ocular volume showed larger results for vitreous volume, as illustrated in Fig 4D. The eye volume, lens volume, anterior chamber volume and vitreous volume of SD rats are respectively about: 0.14±0.02 to 0.15±0.01ml, 0.03±0.00 to 0.03±0.00ml, 0.01±0.00 to 0.01±0.01ml, 0.04±0.01ml. Specific results for both measurement methods are detailed in Table 3.

### Correlation analysis

The gross volumetric estimation of ocular volume results showed a significant positive correlation between the vitreous cavity volume and ocular volume in Bama miniature pigs (p ≤ 0.001, Fig 5A); The vitreous cavity volume of New Zealand rabbits was significantly positively correlated with the ocular volume (p ≤ 0.001), and significantly negatively correlated with the anterior chamber volume (p ≤ 0.05). The real vitreous volume was significantly positively correlated with the lens volume (p ≤ 0.05), as shown in Fig 5B; The vitreous cavity volume of SD rats was significantly positively correlated with the ocular volume (p ≤ 0.001), while the true vitreous volume was significantly negatively correlated with the ocular volume (p ≤ 0.05), as shown in Fig 5C.

As shown in Fig 6A, there was a significant positive correlation (p ≤ 0.001) between the vitreous cavity volume and ocular volume in the CT measurement results of Bama miniature pigs; The eyeball volume of New Zealand rabbits was significantly positively correlated with the vitreous cavity volume (p ≤ 0.001), and also significantly positively correlated with the lens volume (p ≤ 0.05), as shown in Fig 6B; The correlation of CT data in SD rats wasthe same as that in New Zealand rabbits, as shown in Fig 6C.

### Discussion

In this study, we employed micro-CT and anatomical methods to investigate the ocular volume and intraocular contents of Bama miniature pigs, New Zealand rabbits, and SD rats. Our

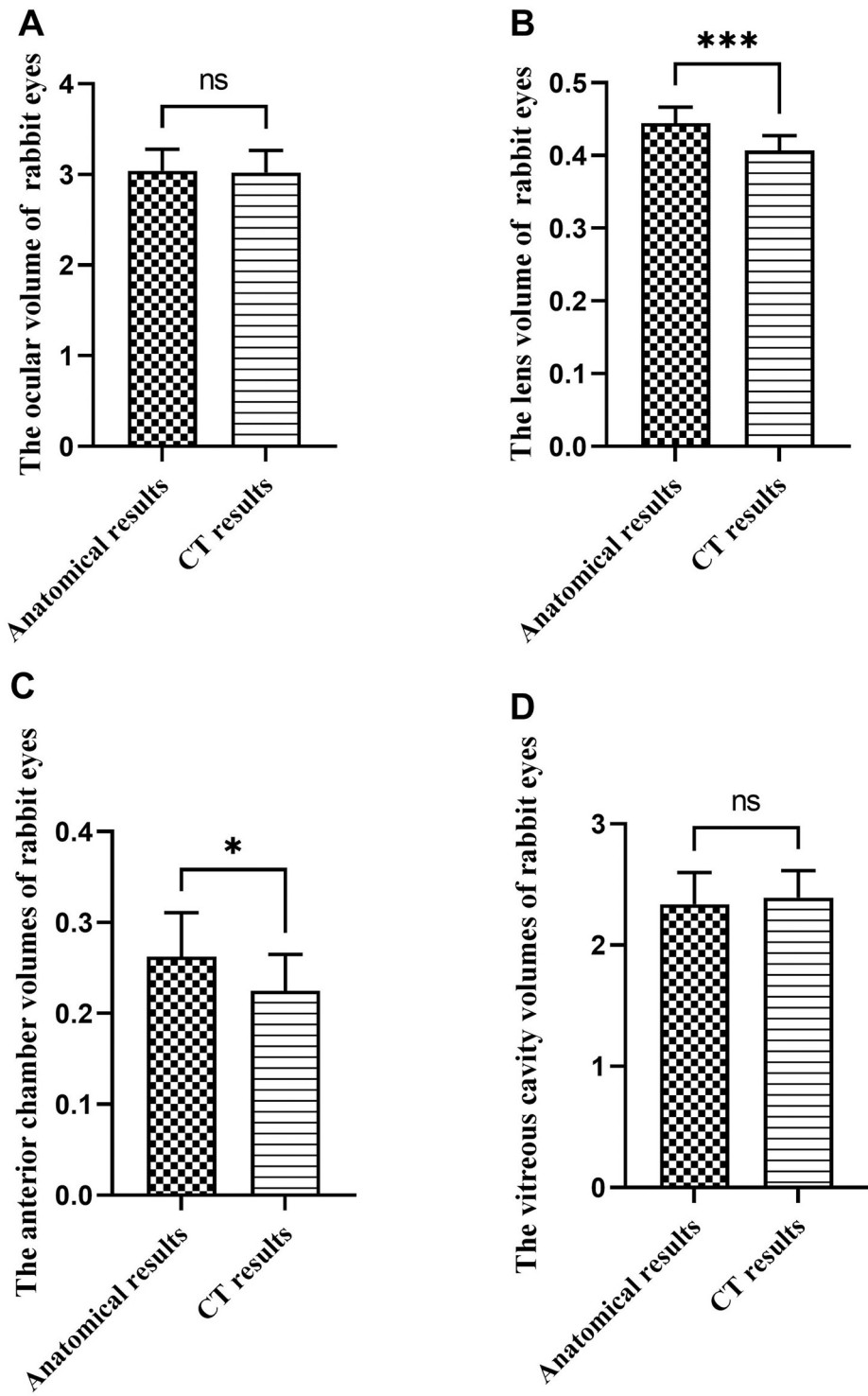

**Fig 3. Comparison of anatomical and CT methods for measuring the eyeball and intraocular volume in New Zealand rabbits (ml).** Note: A, Comparison of two measurement methods in rabbit ocular volume; B, Comparison of two measurement methods in rabbit lens volume; C, Comparison of two measurement methods on the volume of rabbit anterior chamber; D, Comparison of two measurement methods on the volume of rabbit vitreous cavity (The volume of the vitreous cavity here represents the total volume of the vitreous cavity and the remaining eye tissue). ns means that the difference is not statistically significant; *Indicates $P \leq 0.05$; *** Indicates $P \leq 0.001$.

**Table 2. Comparison of anatomical and CT measurements of ocular volumes and intraocular contents volumes in rabbits (ml).**

| Group (n = 12) | Ocular volume | Lens volume | Anterior chamber volume | Vitreous cavity volume (volume of vitreous cavity and remaining eye tissue) | Real vitreous cavity volume |
|---|---|---|---|---|---|
| Gross volumetric estimation of ocular volume | 3.04±0.24 | 0.44±0.02 | 0.26±0.05 | 2.34±0.26 | 1.54±0.14 |
| CT result | 3.02±0.24 | 0.41±0.02 | 0.23±0.04 | 2.39±0.22 | / |
| $t$ value | 0.20 | 4.28 | 2.08 | 0.55 | / |
| $P$ value | 0.84 | $\leq 0.00^{***}$ | $0.05^{*}$ | 0.59 | / |

Note:

*Indicates $P \leq 0.05$;

*** Indicates $P \leq 0.001$.

findings contribute to the understanding of the measurement accuracy and potential applications of these techniques in ophthalmic research. Micro-CT, characterized by high resolution comparable to an optical microscope, provides a non-destructive means of studying internal microstructures [20], allowing for detailed anatomical studies of eye soft tissues [21]. Meanwhile, micro-CT can be applied in animal experiments to live small animals such as rats and rabbits, as well as to study various ex vivo tissue samples. In our study, due to the large size of the pigs, in vivo, eye scanning could not be performed. In order to control the uniform conditions, ex vivo CT scanning was performed on the eyeballs of pigs, rabbits, and rats. Our results revealed no statistically significant difference in ocular volume between CT and gross volumetric estimation of ocular volume across the three animal species. However, deviations in anterior chamber and lens volumes were observed. These discrepancies may arise from challenges in manually delineating the eyeball contour on CT images, potential tissue overlap, and subtle errors in gross volumetric estimation of ocular volume.

Based on the measurement results of the two methods, we observed similarities in pigs and rabbits, indicating that both species exhibited higher anterior chamber and lens volumes measured by anatomical methods compared to micro-CT, with statistically significant differences. Conversely, in SD rats, the anterior chamber volume measured by anatomical methods was smaller than that measured by micro-CT, while the vitreous volume was larger than that measured by CT. Despite the expectation of no difference in the volume of each eyeball content, this was not the case in reality. Compared to SD rats, the eyeballs of Bama miniature pigs and New Zealand rabbits are significantly larger, especially the eyeball volume of pigs, while the eyeballs of New Zealand rabbits are slightly smaller than those of pigs. We identified the main sources of error in obtaining eye content volume in pigs and rabbits using both measurement methods as the anterior chamber volume and lens volume. Micro-CT primarily utilizes scanning and manual delineation to obtain these volumes, reconstructing the anterior chamber and lens comprehensively and calculating their volumes. In contrast, gross volumetric estimation of ocular volume determine the volume of the anterior chamber through a single extraction of aqueous humor and a syringe scale. Therefore, we posit that CT measurement results may be more reliable because only the anterior chamber is delineated in the measurement, whereas the larger volume of the pig and rabbit anterior chamber measured by dissection may be due to the extraction of a small amount of posterior chamber volume.

We euthanized all animals by bleeding the abdominal aorta after anesthesia, and it took less than 2 minutes to completely kill the animals by cutting the abdominal aorta. Our procedure was to quickly transfer the extracted eyeballs into the CT room for scanning, with each eyeball scan lasting approximately 3–5 minutes. The scanned eyeball was first placed into the

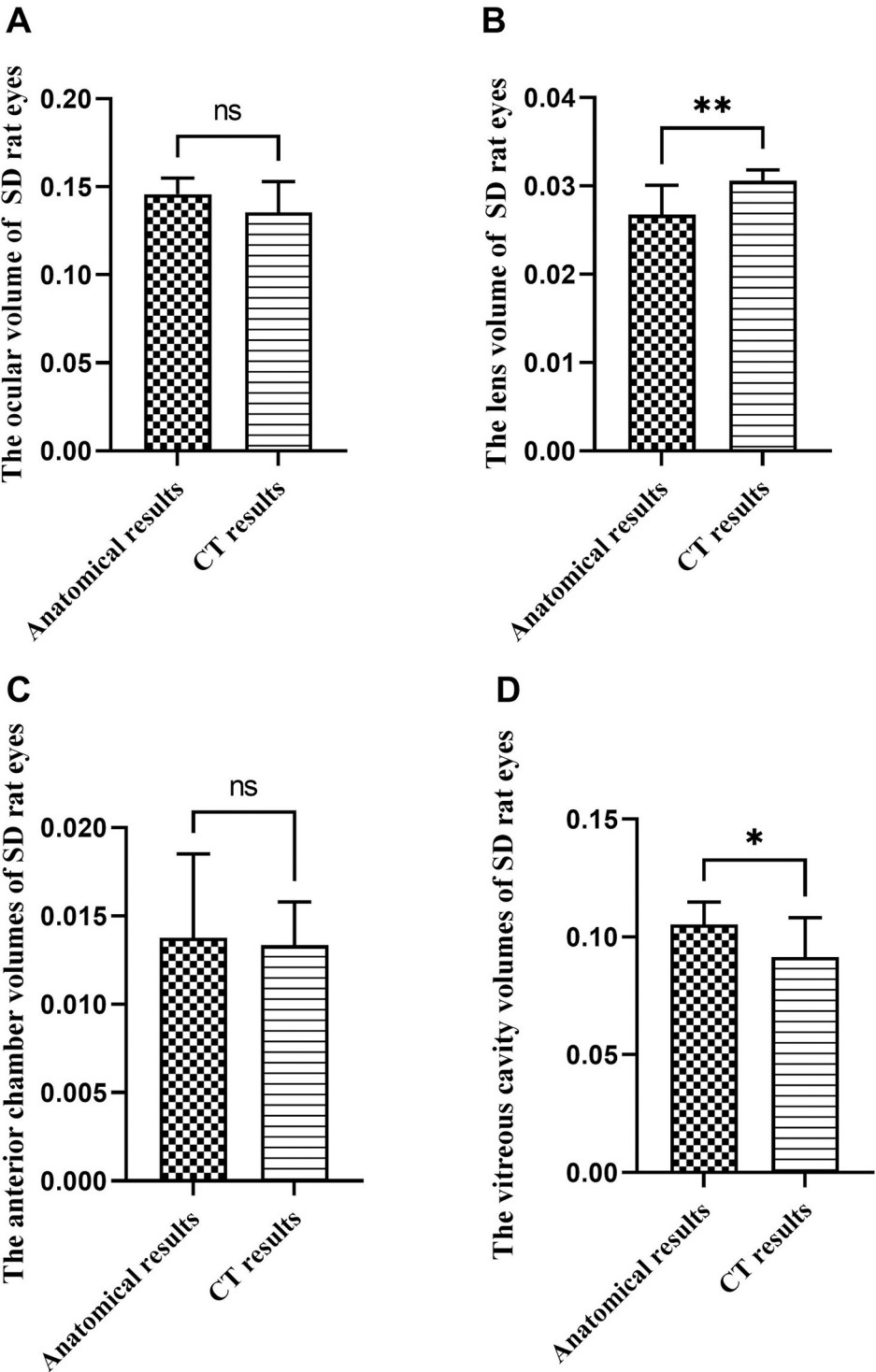

**Fig 4. Comparison of anatomical and CT methods for measuring the eyeball and intraocular volume in SD rats (ml).** Note: A, Comparison of two measurement methods in rat ocular volume; B, Comparison of two measurement methods in rat lens volume; C, Comparison of two measurement methods on the volume of rat anterior chamber; D, Comparison of two measurement methods on the volume of rat vitreous cavity (The volume of the vitreous cavity here represents the total volume of the vitreous cavity and the remaining eye tissue). ns means that the difference is not statistically significant; *Indicates $P \leq 0.05$; ** Indicates $P \leq 0.01$.

**Table 3. Comparison of anatomical and CT measurements of ocular volumes and intraocular contents volumes in SD rats (ml).**

| Group (n = 12) | Ocular volume | Lens volume | Anterior chamber volume | Vitreous cavity volume (volume of vitreous cavity and remaining eye tissue) | Real vitreous cavity volume |
|---|---|---|---|---|---|
| Gross volumetric estimation of ocular volume | 0.15±0.01 | 0.03±0.00 | 0.01±0.01 | 0.11±0.01 | 0.04±0.01 |
| CT result | 0.14±0.02 | 0.03±0.00 | 0.01±0.00 | 0.09±0.02 | / |
| t value | 1.84 | 3.73 | 0.27 | 2.50 | / |
| P value | 0.08 | 0.00** | 0.79 | 0.02* | / |

Note:

*Indicates P ≤ 0.05;

** Indicates P ≤ 0.01.

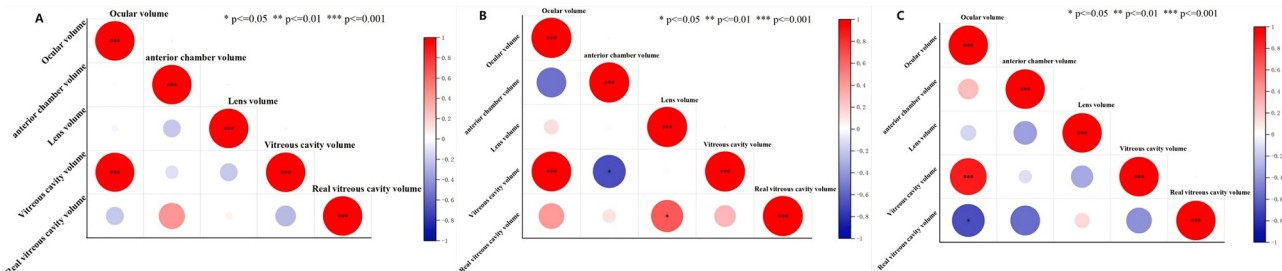

**Fig 5. Correlation heatmap of gross volumetric estimation of ocular volume data.** Note: A. Correlation analysis of Bama miniature pig data; Correlation analysis of New Zealand rabbit data; C. Correlation analysis of SD rat data. Red represents positive correlation, blue represents negative correlation.*P ≤ 0.05; **P ≤ 0.01; ***P ≤ 0.001.

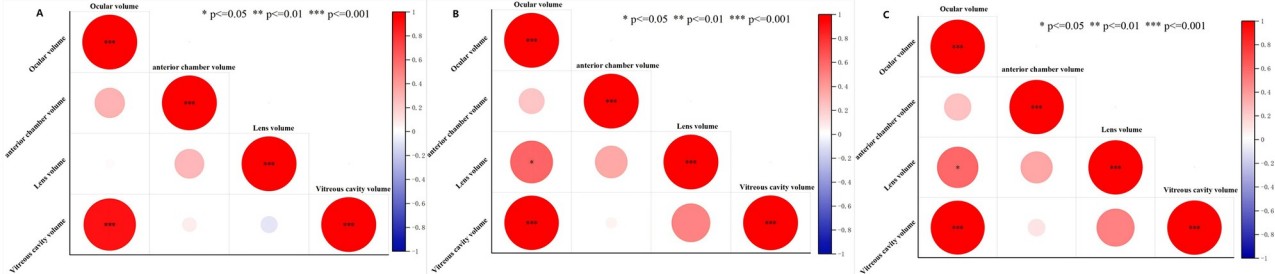

**Fig 6. Correlation heatmap of Correlation heatmap of CT measurement data.** Note: A. Correlation analysis of Bama miniature pig data; Correlation analysis of New Zealand rabbit data; C. Correlation analysis of SD rat data. Red represents positive correlation, blue represents negative correlation. *P ≤ 0.05; **P ≤ 0.01; ***P ≤ 0.001.

anatomical measurement volume for measurement and recording of the eyeball volume, which takes about 1–2 minutes. Then, the aqueous humor was immediately collected, which takes about tens of seconds. The eyeball that has been completely drained of aqueous humor was quickly frozen in liquid nitrogen, and the lens and vitreous body were removed and placed in the anatomical measurement container. After the tissue was thawed, readings were taken. We tried to extract the anterior chamber as dry as possible during the collection of aqueous humor, and evaluate all obtained parameters based on anatomical measurements and CT scans. We mentioned that the entire scanning process was very brief and there was almost no

change in eye size, which is evidenced by the lack of statistical differences in the total eye volume of the three animals in CT and anatomical measurements. Regarding the smaller anterior chamber volume measured by dissection in SD rats compared to CT, SD rats have notably small eyeballs, necessitating higher accuracy in gross volumetric estimation of ocular volume. Additionally, their aqueous humor content is minimal, which may lead to losses during transfer to the measuring instrument. This discrepancy may be the primary reason why the anterior chamber volume measured by dissection is smaller than that measured by CT. The larger vitreous volume is likely due to this volume being mainly the sum of eyeball volumes minus the anterior and lens volumes. A smaller anterior chamber volume results in an overestimation of this portion of the volume. The loss error caused by the larger eyeball structure and contents of rabbits and pigs can be ignored, resulting in no difference between the two methods in vitreous volume.

Furthermore, the lens volumes of pigs and rabbits measured by dissection were both larger than the results of micro-CT. We do not attribute the main source of error to CT, as there is no discrepancy between the eye volume results obtained from CT and gross volumetric estimation of ocular volume without separating the intraocular contents. This suggests that we encounter no significant issues in delineating the eye structure, conducting 3D reconstruction, and automatically measuring volume. We speculate that the main source of error lies in gross volumetric estimation of ocular volume.

The process of anatomical measurement involves completely removing the lens after rapid freezing in liquid nitrogen and then placing it in a container for volume reading after thawing. This process includes freezing, separation, thawing, and data reading, each step potentially introducing errors. Particularly during the separation process, there may be some surrounding tissue included, leading to slightly larger measurement results. However, each lens we obtain remains frozen and intact until completely melted before reading. Additionally, the lens of SD rats is very small, and even if a small amount of surrounding tissue is included, it can be disregarded, resulting in no difference in lens volume between the two measurement methods. We maintain the view that gross volumetric estimation of ocular volume are essential because micro-CT may not clearly distinguish the boundary between the vitreous body and posterior chamber. We rely on gross volumetric estimation of ocular volume to accurately assess the volume of the vitreous body. In reality, we are not comparing two distinct measurement methods, but rather integrating these approaches to derive the volume of the eyeballs and their contents in Bama miniature pigs, New Zealand rabbits, and SD rats. We consider the results obtained from both methods as complementary experimental techniques rather than establishing comparative relationships between them. Therefore, we believe that referencing the results obtained from both methods together is necessary for a comprehensive understanding.

Due to limitations in experimental conditions, the small animal imaging technology available in our laboratory is micro-CT. In fact, possessing micro-CT equipment is considered rare and luxurious for many laboratories. Among the imaging techniques used in ophthalmology, ultrasound imaging (US), CT, and MRI are commonly used. The US uses ultrasound for disease diagnosis, which has the advantages of high soft tissue resolution [22], non-invasive, real-time imaging [23], and low cost [24]. However, its examination accuracy is insufficient, small lesions are easily overlooked, and it does not have a multiplanar imaging function, making it impossible to estimate volume. Besides, MRI has better soft tissue imaging capabilities, enabling 3D stereoscopic imaging, which is non-destructive and radiation-free for patients [25]. Compared with conventional US, MRI can more accurately measure small tumors [26]. However, due to the influence of metals, it cannot be used to measure tissues containing metals, and the operation time is long, with high usage costs and expenses [27, 28]. Compared to the two-dimensional imaging of the US, both CT and MRI, being three-dimensional imaging

techniques, are suitable for volume estimation and are unaffected by metallic elements [29]. The examination time is short, the cost is low, one study found that using abdominal CT examination instead of direct MRI examination for lumbar diseases can save $1.2–3.4 billion per year [30]. And it is also a non-invasive operation [31], although CT radiation is relatively high, it is not significantly affected in animal experiments.

Micro-CT has the same imaging principle as clinical CT [32], but its resolution is higher [33], which is widely used for obtaining animal imaging data [34]. The imaging principle of Micro-CT mainly uses X-rays [35] to pass through the sample and capture hundreds of two-dimensional images from different angles around the sample. The software can reconstruct images from various angles to generate three-dimensional images [36], and it can also outline the parts of interest in the sample, and even create 3D animations. In addition, micro-CT can measure the size, volume, and spatial coordinates of each point of the sample, as well as the attenuation value and density information of the sample. In the study, we used micro-CT to scan the eyeballs of Bama miniature pigs, New Zealand rabbits, and SD rats. We manually delineated the anterior chamber and lens using software to construct complete stereoscopic images of the eyeballs, and successfully obtained the volumes of various regions of interest. However, the gap between the vitreous cavity and the posterior chamber on CT images is unclear, making it difficult to manually sketch the vitreous cavity, which is also one of the reasons why we combined gross volumetric estimation of ocular volume.

In fact, in order to prevent the unclear structure of the eyeball during the experimental design, we wanted to enhance image contrast by combining staining and micro-CT scanning. Like Leszczyński B, et al. [14] stained the eyeballs of domestic pigs with iodine, 100% Lugol, phosphotungstic acid, and 1% osmium tetroxide solutions, and then performed micro-CT scans. They found that staining can improve the visualization of eyeball structure, and different staining agents and staining time can affect the imaging results. However, the eyeball of SD rats is very small, and we are afraid that injecting staining agents may increase the intraocular volume. Therefore, we decided to dissect the eyeball after CT scanning separately to obtain its anatomical data, and estimate the volume of the eyeball and its contents by combining CT and gross volumetric estimation of ocular volume. Notably, the combination of micro-CT and gross volumetric estimation of ocular volume in this study allowed for comprehensive data acquisition, although challenges were encountered in delineating the vitreous cavity due to unclear boundaries. Future improvements could involve enhancing image contrast through staining. Our study lays the foundation for estimating eye tissue volumes in Bama miniature pigs, New Zealand rabbits, and SD rats, offering insights for drug research and intraocular drug volume conversion in these experimental animals.

Our correlation analysis results indicated a significant positive correlation (p ≤ 0.001) between the ocular volume and the vitreous cavity volume, regardless of anatomical or CT measurements, this may be mainly because the volume of the vitreous cavity was the sum of the volume of the real vitreous cavity and the remaining eye tissue, which was closer to the volume of the eyeball. But in the results of gross volumetric estimation of ocular volume. For example, we found that the vitreous cavity volume of New Zealand rabbits had a significant negative correlation with the anterior chamber volume, while the real vitreous cavity volume had a significant positive correlation with the lens volume. There was a significant negative correlation between the true vitreous cavity volume and eyeball volume in SD rats, which did not exist in the data of pig eyeball. We speculated that this might be due to the difference in the proportion of different eye contents in pigs, rabbits and SD rats (The lens of SD rats accounts for 18% of the total volume of the eyeball, while the true vitreous cavity accounts for 28%; the crystalline lens of rabbits accounts for 15% of the total volume of the eyeball, while the true vitreous cavity accounts for 50%; and the crystalline lens of pigs accounts for 6% of the total

volume of the eyeball, while its true vitreous cavity accounts for 58%.). However, due to imaging defects in micro-CT, CT measurements cannot obtain the true volume of the vitreous cavity, so the correlation analysis results between the three different animal eye parameters measured by CT were almost identical.

So far, no other studies have reported on the volume of eyeballs and their contents in various structures of Bama miniature pigs, New Zealand rabbits, or SD rats using combined anatomical and imaging measurement methods, as observed in our study. However, some studies have utilized micro-CT to assess the volume of intraocular structures in animals. For instance, Atwood RC et al. [37] employed three-dimensional reconstruction via micro-CT and computer-aided image analysis technology to quantify the volume of intraocular vessels in mice. Similarly, Leszczynski et al. [14] successfully utilized micro-CT to obtain data on pig eyeball volume. Diverging from the aforementioned studies, which focused on quantifying the volume of eye contents at specific sites, our study aimed to elucidate the volume of eyeballs and their contents across several commonly used experimental animals. This approach aimed to provide practical reference values for the utilization of relevant experimental drugs and the conversion of drug volume between species. Therefore, we conducted comprehensive measurements of the volume of the anterior chamber, lens, and vitreous body. Nevertheless, our research is built upon the foundation laid by previous studies, which affirmed and elucidated the value of micro-CT and its 3D reconstruction technology in measuring eyeball volume. These earlier investigations paved the way for our study's methodology and its contributions to the field.

Despite its contributions, our study has several limitations. Firstly, measurement errors, particularly inherent to the traditional anatomical measurement methods we employed, remain a challenge. Despite our efforts to enhance the accuracy of each measurement step, errors, especially in measuring anterior chamber and lens volume, are inevitable. We anticipate that future advancements in measurement technology may mitigate these issues. Furthermore, the indistinct boundary between the vitreous cavity and posterior chamber in micro-CT images presents a significant challenge. While we attempted to optimize imaging parameters to achieve clearer images, none proved successful. Additionally, we hesitated to use staining techniques due to concerns about increasing intraocular volume. Access to equipment with clearer resolution, such as micro-MRI or high-resolution micro-CT, could potentially offer solutions. However, our current laboratory lacks the resources to acquire such equipment.

## Conclusion

Our study demonstrates the feasibility of estimating the volume of the eyeball and its contents through a combined approach utilizing micro-CT and anatomical methods in Bama miniature pigs, New Zealand rabbits, and SD rats. Notably, we observed variations in the proportion of eye contents relative to the eyeball volume across different species. Specifically, SD rats exhibited the largest proportion of lens volume in the total eyeball volume, emphasizing the potential vulnerability of the lens during intraocular procedures in this species. Further refinements and technological advancements are crucial for enhancing precision and expanding the applications of these techniques in ophthalmic research.

## Supporting information

**S1 Data.**
(ZIP)

**S2 Data.**
(ZIP)

**S3 Data.**
(ZIP)

**S4 Data.**
(ZIP)

## Acknowledgments

Thanks to WestChina-Frontier PharmaTech Co., Ltd. for providing laboratory facilities and animals for this study.

## Author Contributions

**Data curation:** Jiasong Yang.

**Formal analysis:** Zongtao Shu.

**Investigation:** Jiasong Yang, Yuwen Ran.

**Methodology:** Yuwen Ran, Zongtao Shu.

**Project administration:** Xiaobo Cen, Wensheng Li.

**Resources:** Xiaobo Cen.

**Supervision:** Wensheng Li.

**Visualization:** Yuwen Ran.

**Writing – original draft:** Yajun Wu.

**Writing – review & editing:** Yajun Wu, Yuliang Feng.

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
