## [Decision Letter · Decision Letter 0]

27 Aug 2024

PONE-D-24-29172Anatomical and Micro-CT measurement analysis of ocular volume and intraocular volume in adult Bama Miniature Pigs, New Zealand rabbits, and Sprague-Dawley ratsPLOS ONE

Dear Dr. Li,

Thank you for submitting your manuscript to PLOS ONE. After careful consideration, we feel that it has merit but does not fully meet PLOS ONE’s publication criteria as it currently stands. Therefore, we invite you to submit a revised version of the manuscript that addresses the points raised during the review process.

Based on the Reviewers` comment the current manuscript needs some minor revision. Please make a correction of the paper according to these suggestions.

We look forward to receiving your revised manuscript.

Kind regards,

Karolina Goździewska-Harłajczuk

Academic Editor

PLOS ONE

“2023 Research Fund of Aier Eye Research Institute(No.AEI202310LC01); Science Research Foundation of Aier Eye Hospital Group (No.AR2201D3).”

“Thanks to Aier Eye Group for its funding support; Thanks to WestChina-Frontier PharmaTech Co., Ltd. for providing laboratory facilities and animals for this study.”

“2023 Research Fund of Aier Eye Research Institute(No.AEI202310LC01); Science Research Foundation of Aier Eye Hospital Group (No.AR2201D3).”

4. In the online submission form you indicate that your data is not available for proprietary reasons and have provided a contact point for accessing this data. Please note that your current contact point is a co-author on this manuscript. According to our Data Policy, the contact point must not be an author on the manuscript and must be an institutional contact, ideally not an individual. Please revise your data statement to a non-author institutional point of contact, such as a data access or ethics committee, and send this to us via return email. Please also include contact information for the third party organization, and please include the full citation of where the data can be found.

Additional Editor Comments:

Based on the Reviewers` comment the current manuscript needs some minor revision.

Reviewers' comments:

Reviewer's Responses to Questions

**Comments to the Author**

1. Is the manuscript technically sound, and do the data support the conclusions?

Reviewer #1: Yes

Reviewer #2: Yes

2. Has the statistical analysis been performed appropriately and rigorously? 

Reviewer #1: Yes

Reviewer #2: I Don't Know

3. Have the authors made all data underlying the findings in their manuscript fully available?

Reviewer #1: Yes

Reviewer #2: Yes

4. Is the manuscript presented in an intelligible fashion and written in standard English?

Reviewer #1: Yes

Reviewer #2: Yes

5. Review Comments to the Author

Reviewer #1: The study aimed to measure the eyeball and its components in Bama Miniature Pigs, New Zealand rabbits, and Sprague-Dawley rats using micro-CT and anatomical techniques. They included six animals from each species and measured the volumes of the eyeball, anterior chamber, lens, and vitreous cavity. The conclusion was that combining micro-CT and anatomical measurements is effective for determining eyeball volumes in these animals. This information could be useful for future ophthalmology-related drug research.

Reviewer #2: Review on the manuscript PONE-D-24-29172

Anatomical and Micro-CT measurement analysis of ocular volume and intraocular

volume in adult Bama Miniature Pigs, New Zealand rabbits, and Sprague-Dawley rats

Thanks very much for giving me such opportunity to revise the current manuscript. The manuscript describes the anatomical and micro-CT examination of the eye in Bama miniature pigs, New Zealand rabbits and Sprague-Dawley rats.

The manuscript could be considered for publication. My main concern is that the study design is not clear, was it an in vivo or ex vivo or in vitro (as indicated by authors). It is not clear when animals really euthanized (before or after CT scan). If before scanning, how did the author maintain accuracy of measurements following euthanasia?

Additionally, the method of euthanasia requires justification.

Minor comments:

Title:

- Replace “Pig” with “pig”.

Abstract:

- You can report your results using 2 decimal points.

Introduction:

- Line 78: “Salguero R et al.”…replace with “Salguero et al.”

- “length of each line in normal dog”….explain what do you mean by each line.

Method:

- Lone 89: “This is an in vitro experiment”…..Do you believe that this is an in vitro experiment?! ….. please replace by “ an in vivo study”.

- Line 103: “The tool was then used to draw the ROI (region of interest)” could be replaced by “The tool was then used to draw the region of interest (ROI)”

- Line 110: “All operations were performed by a single experimenter” … Do you mean “All CT scans and measurements were made by the same examiner (add Name abbreviations)?? ….Did you made measurements at the same time of examination or you made it on saved scans….please clarify. Did you make your measurements only one time or multiple times and get your mean for each eye measurements.

- Line 111: Anatomical results/anatomic measurements all over the text could be replaced by “Gross volumetric estimation of ocular volume”

- Line 112-113:

- “intravenous injection of pentobarbital sodium (30mg/kg)” at which vein did you inject in each animal?

- please add concentration, trade name, manufacturer and country of origin of the anesthetic drugs.

- “all animals were euthanized through abdominal artery bleeding” ….Which abdominal artery? How did you approach it? What is your justification and reference for this method of euthanasia? How did you ensure euthanasia?

Statistical analysis:

- Did you consider running correlation and regression analysis to your measurements?

Results:

- Results could be presented using 2 decimal points.

Discussion:

- Line 162-163: “In our study, due to the large size of the pigs, in vivo, eye scanning could not be performed. In order to control the uniform conditions, ex vivo CT scanning was performed on the eyeballs of pigs, rabbits, and rats”…..this is really very confusing. For the first time here in the discussion section to include that you did not make your CT scans on life animals??!!! You have mentioned that animals were euthanized for the anatomic study not for the CT scans.

Then how did you position the eyes balls for scanning? How did you manage post-mortem changes of ocular dimensions during scanning??

Please clarify in details.

Line 173: you are suggesting that the micro-CT measurements are more reliable than anatomic measurements “we posit that CT measurement results may be more reliable”……So what was your study objective and hypothesis??!! What is your gold standard to judge/prove that micro CT is reliable in determination of volumetric eye measurement?

Figures:

- Figure 1: please add reference to panels a,b, c within “Figure caption”. You may need to include description of volumetric measurements in the CT scan.

Thanks

6. PLOS authors have the option to publish the peer review history of their article (what does this mean?). If published, this will include your full peer review and any attached files.

Reviewer #1: **Yes: **Rodrigo Pessoa Cavalcanti Lira

Reviewer #2: No

---

## [Author Response · Author response to Decision Letter 0]

31 Aug 2024

Dear Editor and reviewers, thank you very much for giving us the opportunity to revise this manuscript. We have carefully revised the manuscript according to your requirements, and the relevant response is as follows. Once again, we sincerely thank you!

To academic editor and journal:

1.Please ensure that your manuscript meets PLOS ONE's style requirements, including those for file naming. 

Reply:

Dear editor, thank you again for providing us with this opportunity to revise the manuscript. We have changed the revised manuscript into the correct format as required by the journal, please check it!

“2023 Research Fund of Aier Eye Research Institute(No.AEI202310LC01); Science Research Foundation of Aier Eye Hospital Group (No.AR2201D3).” Please state what role the funders took in the study. If the funders had no role, please state: "The funders had no role in study design, data collection and analysis, decision to publish, or preparation of the manuscript." If this statement is not correct you must amend it as needed. Please include this amended Role of Funder statement in your cover letter; we will change the online submission form on your behalf.

Reply:

Dear editor, thank you very much for your comment, we are very sorry for not stating the role of funders in the cover letter, we have added relevant information in the cover letter, thank you very much for your comments again!

3.Thank you for stating the following in the Acknowledgments Section of your manuscript:

“Thanks to Aier Eye Group for its funding support; Thanks to WestChina-Frontier PharmaTech Co., Ltd. for providing laboratory facilities and animals for this study.”We note that you have provided funding information that is currently declared in your Funding Statement. However, funding information should not appear in the Acknowledgments section or other areas of your manuscript. We will only publish funding information present in the Funding Statement section of the online submission form.

Reply:

Dear editor, thank you very much for your comment. We have deleted the funding information from the manuscript and added it in the cover letter, please check it!

4.In the online submission form you indicate that your data is not available for proprietary reasons and have provided a contact point for accessing this data. Please note that your current contact point is a co-author on this manuscript. According to our Data Policy, the contact point must not be an author on the manuscript and must be an institutional contact, ideally not an individual. Please revise your data statement to a non-author institutional point of contact, such as a data access or ethics committee, and send this to us via return email. Please also include contact information for the third party organization, and please include the full citation of where the data can be found.

Reply:

Dear editor, thank you very much for your comment. We wanted to submit the original data when submitting the manuscript, but the original data was too large to be submitted. That's why we wrote in the data statement that we can obtain the original data from the corresponding author. In this revised draft, we have split the original data and uploaded it, please check, thank you!

5.Please review your reference list to ensure that it is complete and correct. If you have cited papers that have been retracted, please include the rationale for doing so in the manuscript text, or remove these references and replace them with relevant current references. Any changes to the reference list should be mentioned in the rebuttal letter that accompanies your revised manuscript. If you need to cite a retracted article, indicate the article’s retracted status in the References list and also include a citation and full reference for the retraction notice.

Reply:

Dear editor, thank you very much for your comment. The references cited in this manuscript are complete and correct, and there are no references to the retracted manuscript.

To additional editor:

Based on the Reviewers` comment the current manuscript needs some minor revision.

Reply:

Dear editor, thank you again for providing us with this opportunity to revise the manuscript. We have made corresponding revisions to the manuscript as required by the reviewers, please check it! Thanks again for this opportunity！

To reviewers:

To Reviewer #1: 

The study aimed to measure the eyeball and its components in Bama Miniature Pigs, New Zealand rabbits, and Sprague-Dawley rats using micro-CT and anatomical techniques. They included six animals from each species and measured the volumes of the eyeball, anterior chamber, lens, and vitreous cavity. The conclusion was that combining micro-CT and anatomical measurements is effective for determining eyeball volumes in these animals. This information could be useful for future ophthalmology-related drug research.

Reply:

Dear reviewer, thank you very much for your detailed review of this manuscript. Your comments are an important recognition and support for our work. Thank you again for your comments.

To Reviewer #2: 

1.Title:

- Replace “Pig” with “pig”.

Abstract:

- You can report your results using 2 decimal points.

Introduction:

- Line 78: “Salguero R et al.”…replace with “Salguero et al.”

- “length of each line in normal dog”….explain what do you mean by each line.

Reply:

Dear reviewer, thank you very much for your comments. We have made corresponding modifications in the manuscript. Besides, we are very sorry for the confusion caused by our writing, the "length of each line in normal dog" refers to the distance between certain structures of the dog's eyeball, including the average axial length of the sphere, the average anterior posterior distance of the anterior chamber, the average anterior posterior distance of the vitreous chamber, etc. For better understanding, we have replaced this sentence with “and the length of the normal dog eye structures (Including the average axial length of the sphere, the average anterior posterior distance of the anterior chamber, the average anterior posterior distance of the vitreous chamber, etc) with CT”, please check it.

2.Method:

- Lone 89: “This is an in vitro experiment”…..Do you believe that this is an in vitro experiment?! ….. please replace by “ an in vivo study”.

- Line 103: “The tool was then used to draw the ROI (region of interest)” could be replaced by “The tool was then used to draw the region of interest (ROI)”

Reply:

Dear reviewer, thank you very much for your comments. We have made corresponding modifications in the manuscript. Besides, due to our inability to perform micro CT scans on live pigs and rabbits, the scanning chamber for small animal CT is very small and can only accommodate mice and rats. In order to standardize the measurement method, we removed the eyeball and conducted separate CT and anatomical measurements. We euthanize animals in order to extract their eyeballs for measurement of in vitro parameters. Strictly speaking, we should not consider it as an in vivo experiment. To avoid misunderstandings, we have removed the mention of in vivo and in vitro experiments. We have provided corresponding descriptions of the measurement methods in our methodology.please check it.

3.Line 110: “All operations were performed by a single experimenter” … Do you mean “All CT scans and measurements were made by the same examiner (add Name abbreviations)?? ….Did you made measurements at the same time of examination or you made it on saved scans….please clarify. Did you make your measurements only one time or multiple times and get your mean for each eye measurements.

Reply:

Dear reviewer, thank you very much for your comment. In order to reduce errors, our CT scans and eye measurements under CT were performed by the same professional imaging technician (Zongtao Shu). Due to the time-consuming process of manual delineation and reconstruction of eye structure involved in the measurement process, our technicians promptly saved all eye image data after scanning, and then uniformly measured eye volume. In addition, we also referred to other literature and did not have similar imaging data for multiple measurements. Therefore, the CT measurement data for this part was only reconstructed by outlining the eyeball structure once to obtain the required parameter results. The entire process took nearly 2 months and was quite cumbersome and complex.

4.Line 111: Anatomical results/anatomic measurements all over the text could be replaced by “Gross volumetric estimation of ocular volume”.

Reply:

Dear reviewer, thank you very much for your comments. We have made corresponding modifications in the manuscript, please check it.

5.- Line 112-113:

- “intravenous injection of pentobarbital sodium (30mg/kg)” at which vein did you inject in each animal?

- please add concentration, trade name, manufacturer and country of origin of the anesthetic drugs.

Reply:

Dear reviewer, thank you for your detailed comments on this manuscript. Both Bama miniature pigs and New Zealand rabbits were injected via ear vein after skin preparation, while SD rats were injected intraperitoneally with 30mg/kg pentobarbital sodium. We are very sorry that we did not provide detailed information on the concentration, product name, manufacturer, and country of origin of the anesthetic drug in the manuscript. Our reagents are 2mg/kg diazepam (Sigma-Aldrich, USA) and 30mg/kg pentobarbital sodium (Sigma-Aldrich, USA), which we have supplemented in the manuscript. Please check. Thank you!

6.“all animals were euthanized through abdominal artery bleeding” ….Which abdominal artery? How did you approach it? What is your justification and reference for this method of euthanasia? How did you ensure euthanasia?

Reply:

Dear reviewer, thank you very much for your comments. All animals in this manuscript were euthanized with diazepam and anesthetized with 30mg/kg pentobarbital sodium before cutting the abdominal aorta and bleeding to death. We have supplemented this information in the manuscript. We ensure that all animals have a safe, effective, painless, and pain free death experience, following the AVMA Animal Euthanasia Guidelines. In addition, we confirmed whether the signs of life had completely disappeared by observing the disappearance of the animal's breathing and heartbeat, as well as pinching its toe reflex. After confirming that the animal had completely died, we proceeded to remove the eyeball, which we also supplemented in the manuscript. Finally, thank you again for your comments!

7.Statistical analysis:

- Did you consider running correlation and regression analysis to your measurements?

Reply:

Dear reviewer, thank you very much for your valuable comment. We have added relevant statistics to the manuscript, please check it!

8.Results:

- Results could be presented using 2 decimal points.

Reply:

Dear reviewer, thank you very much for your comments. We have made corresponding modifications in the manuscript, please check it.

9.Discussion:

- Line 162-163: “In our study, due to the large size of the pigs, in vivo, eye scanning could not be performed. In order to control the uniform conditions, ex vivo CT scanning was performed on the eyeballs of pigs, rabbits, and rats”…..this is really very confusing. For the first time here in the discussion section to include that you did not make your CT scans on life animals??!!! You have mentioned that animals were euthanized for the anatomic study not for the CT scans.

Then how did you position the eyes balls for scanning? How did you manage post-mortem changes of ocular dimensions during scanning??

Please clarify in details.

Reply:

Dear reviewer, thank you very much for your valuable comments. We are very sorry that our manuscript has caused you inconvenience. In fact, we have written about the specific process in the measurement of eye volume in methodology: we euthanize all animals by bleeding the abdominal aorta after anesthesia, and it takes less than 2 minutes to completely kill the animals after cutting the abdominal aorta. Our procedure is to quickly transfer the extracted eyeballs into the CT room for scanning, with each eyeball scan lasting approximately 3-5 minutes. The scanned eyeball is first placed into the anatomical measurement volume for measurement and recording of the eyeball volume, which takes about 1-2 minutes. Then, the aqueous humor is immediately collected, which takes about tens of seconds. The eyeball that has been completely drained of aqueous humor is quickly frozen in liquid nitrogen, and the lens and vitreous body are removed and placed in the anatomical measurement container. After the tissue is thawed, readings are taken. We try to extract the anterior chamber as dry as possible during the collection of aqueous humor, and evaluate all obtained parameters based on anatomical measurements and CT scans.

 We are unable to perform micro-CT scans on live pigs and rabbits because the scanning chamber for small animal-CT is very small and can only accommodate mice and rats. In order to standardize the measurement method, we removed the eyeball and conducted separate CT and anatomical measurements. We euthanize animals in order to extract their eyeballs for measurement of in vitro parameters. Strictly speaking, we should not consider it as an in vivo experiment. To avoid misunderstandings, we have removed the mention of in vivo and in vitro experiments. We have provided corresponding descriptions of the measurement methods in our methodology. In addition, we mentioned that the entire scanning process was very brief and there was almost no change in eye size, which is evidenced by the lack of statistical differences in the total eye volume of the three animals in CT and anatomical measurements.

 Sorry, we did not provide a detailed explanation of the specific method for locating the eyeball for scanning. We have added it to the methodology (The specific way we locate the eyeball for scanning is to ensure that the entire eyeball is located within the scanning interval, manually select the lens to start drawing from the tip of the ciliary body, move towards the other end until the lens structure disappears, and then use this tool to draw the region of interest (ROI) of the lens until the anatomical structure disappears; Using the same method, select any starting surface of the eyeball or anterior chamber structure and draw its ROI until the anatomical structure disappears,the vitreous body and posterior chamber cannot be located due to the lack of a clear boundary line), please check it. Finally, thank you again for your valuable comments. Thank you!

10.Line 173: you are suggesting that the micro-CT measurements are more reliable than anatomic measurements “we posit that CT measurement results may be more reliable”……So what was your study objective and hypothesis??!! What is your gold standard to judge/prove that micro CT is reliable in determination of volumetric eye measurement?

Reply:

Dear reviewer, thank you very much for your detailed review of this manuscript. We are pleased with your professional feedback. We are very sorry that we have not actually recognized micro CT as the gold standard for measuring eye volume. We have also mentioned in our discussion that MRI is actually a more accurate measurement method, but small animal MRI is a very luxurious existence in our laboratory, and currently we do not have the conditions to implement it. The main reason why we believe CT results are more reliable is as follows: we found that there were differences between the anterior chamber volume of the eyeballs of three animals and the CT results. For example, the anatomical measurement of the anterior chamber volume in pigs was 0.28±0.04ml, while the CT result was 0.19±0.05ml. We can only combine the results of the two methods to conclude that the anterior chamber volume in pigs is approximately 0.19±0.05-0.28±0.04ml. We are not sure if time will have a certain impact on the amount of room water, but there are indeed differences between these two measurement methods on room water. In fact, we believe that CT measurement results are more reliable becaus

---

## [Decision Letter · Decision Letter 1]

8 Sep 2024

Anatomical and Micro-CT measurement analysis of ocular volume and intraocular volume in adult Bama Miniature pigs, New Zealand rabbits, and Sprague-Dawley rats

PONE-D-24-29172R1

Dear Dr. Li,

We’re pleased to inform you that your manuscript has been judged scientifically suitable for publication and will be formally accepted for publication once it meets all outstanding technical requirements.

Kind regards,

Karolina Goździewska-Harłajczuk

Academic Editor

PLOS ONE

Additional Editor Comments (optional):

All comments of the Reviewers` were included in the revised version of the mauscript, thus the paper can be accept in current form.

Reviewers' comments:

Reviewer's Responses to Questions

**Comments to the Author**

1. If the authors have adequately addressed your comments raised in a previous round of review and you feel that this manuscript is now acceptable for publication, you may indicate that here to bypass the “Comments to the Author” section, enter your conflict of interest statement in the “Confidential to Editor” section, and submit your "Accept" recommendation.

Reviewer #2: All comments have been addressed

2. Is the manuscript technically sound, and do the data support the conclusions?

Reviewer #2: Yes

3. Has the statistical analysis been performed appropriately and rigorously? 

Reviewer #2: Yes

4. Have the authors made all data underlying the findings in their manuscript fully available?

Reviewer #2: Yes

5. Is the manuscript presented in an intelligible fashion and written in standard English?

Reviewer #2: Yes

6. Review Comments to the Author

Reviewer #2: Thanks very much for giving me such opportunity to revise the current version of the manuscript. The manuscript was greatly improved compared to previous version. Authors are acknowledged for their efforts in responding to previous comments. The manuscript is acceptable for publication.

7. PLOS authors have the option to publish the peer review history of their article (what does this mean?). If published, this will include your full peer review and any attached files.

Reviewer #2: **Yes: **Elham A. Hassan

---

## [Editor Report · Acceptance letter]

11 Sep 2024

PONE-D-24-29172R1 

PLOS ONE

Dear Dr. Li, 

I'm pleased to inform you that your manuscript has been deemed suitable for publication in PLOS ONE. Congratulations! Your manuscript is now being handed over to our production team.

Kind regards, 

on behalf of

Dr. Karolina Goździewska-Harłajczuk 

Academic Editor

PLOS ONE